# Skeletal Muscle in ALS: An Unappreciated Therapeutic Opportunity?

**DOI:** 10.3390/cells10030525

**Published:** 2021-03-02

**Authors:** Silvia Scaricamazza, Illari Salvatori, Alberto Ferri, Cristiana Valle

**Affiliations:** 1Fondazione Santa Lucia IRCCS, c/o CERC, 00143 Rome, Italy; silviascaricamazza@gmail.com (S.S.); i.salvatori@hsantalucia.it (I.S.); 2Department of Biology, University of Rome Tor Vergata, 00133 Rome, Italy; 3Department of Experimental Medicine, University of Rome “La Sapienza”, 00161 Rome, Italy; 4Institute of Translational Pharmacology, National Research Council, 00133 Rome, Italy

**Keywords:** amyotrophic lateral sclerosis, skeletal muscle, pharmacological approaches, physical activity, genetic intervention

## Abstract

Amyotrophic lateral sclerosis (ALS) is a neurodegenerative disorder characterized by the selective degeneration of upper and lower motor neurons and by the progressive weakness and paralysis of voluntary muscles. Despite intense research efforts and numerous clinical trials, it is still an incurable disease. ALS had long been considered a pure motor neuron disease; however, recent studies have shown that motor neuron protection is not sufficient to prevent the course of the disease since the dismantlement of neuromuscular junctions occurs before motor neuron degeneration. Skeletal muscle alterations have been described in the early stages of the disease, and they seem to be mainly involved in the “dying back” phenomenon of motor neurons and metabolic dysfunctions. In recent years, skeletal muscles have been considered crucial not only for the etiology of ALS but also for its treatment. Here, we review clinical and preclinical studies that targeted skeletal muscles and discuss the different approaches, including pharmacological interventions, supplements or diets, genetic modifications, and training programs.

## 1. Introduction

Amyotrophic lateral sclerosis (ALS) is a progressive disease characterized by motor neuron degeneration and skeletal muscle atrophy [1]. There are no effective therapies for ALS: riluzole and edaravone are the only FDA-approved drugs; however, they have a rather modest impact on the course of the disease [2].

While the majority of ALS cases are sporadic (sALS), about 10% of the cases are familial (fALS) and characterized by autosomal dominant inheritance. The clinical manifestations of sALS and fALS are indistinguishable, suggesting that different pathways converge, causing the typical neuromuscular degeneration of ALS [3]. However, despite the intense efforts to identify the pathogenetic mechanisms, the etiology of ALS remains elusive.

As ALS has long been considered the prototype of motor neuron diseases, many studies on its pathology have been “neurocentric”. However, since the early 2000s, several papers have started describing the key roles of non-neuronal cell types in triggering or supporting the ALS neuromuscular degenerative processes [4]. Today, ALS is considered a multisystemic and multifactorial disease characterized by the degeneration of motor neurons in the motor cortex and spinal cord and accelerated by physiological alterations of other cell types and organs [5].

Genetic models showed that only the ubiquitous expression of a gain-of-function mutant of the human superoxide dismutase 1 (mSOD1) induced fast and severe paralysis in mice, mimicking ALS progression [6]. The overexpression of mSOD1 in neurons led to a late-onset of the disease with slow progression [7], while restricting the expression of mSOD1 to motor neurons did not trigger the pathology [8]. mSOD1 expression in skeletal muscle elicited muscle atrophy, decreased muscle strength, impaired mitochondrial distribution and the contractile apparatus [9], reduced the spinal cord mass, triggered late motor neuron loss, and shortened the lifespan [10].

The neuromuscular junction (NMJ) connects muscle fibers and motor neurons, allowing their communication; ALS is the classic example of severely compromised communication between muscles and nerves [11]. Motor neuron activity regulates muscle physiology and function; in turn, muscles affect the neuronal activity by sending retrograde signals that preserve NMJ functionality and structure [12]. The so-called “dying back” hypothesis suggests that retrograde signals contribute to the centripetal motor neuron degeneration in ALS [13]. Studies in ALS mouse models have corroborated this hypothesis and described ALS as a distal axonopathy also caused by alterations in skeletal muscle [14].

Different studies have reported that more than 60% of familial and sporadic ALS patients have increased resting and non-resting energy expenditure, an unexpected feature considering the effects of undernutrition on energy balance and the defense mechanisms to lower energy waste [15,16,17]. A study involving a large cohort of ALS patients found that high levels of physical activity were related to an increased risk of ALS [18,19]. It also has been shown that a low premorbid body mass index (BMI) increases the risk of ALS and that weight loss and hypermetabolism correlate with a less favorable prognosis [16,20]. Interestingly, variants of the ACSL5 gene, previously associated with rapid weight loss in humans, have recently been associated with ALS risk and lean body mass reduction in ALS patients [21]. Moreover, ALS incidence is lower among obese individuals, and patients with a high pre-diagnostic body and subcutaneous fat have a lower mortality risk [22,23]. Overall, the studies suggest a key role of energy expenditure and hypermetabolism in ALS pathology.

The cause of defective energy homeostasis in ALS is still unknown; however, since muscle metabolism is the major determinant of the total energy expenditure, we should further investigate the role of skeletal muscle in ALS etiology to understand whether the metabolic disorder contributes to its pathogenesis. Early events before denervation affecting muscle physiology have been described, supporting the “dying back” hypothesis in ALS. Understanding the molecular mechanisms involved in skeletal muscle degeneration may help develop therapeutic strategies that preserve muscle function, slow down the disease progression, and improve ALS patients’ quality of life.

In this review, we have summarized and discussed the therapeutic approaches that have been used to increase the performance of skeletal muscle in ALS animal models and patients.

## 2. Genetic Interventions

Gene therapy aims to treat a disease by inserting genetic material into cells with viral or non-viral vectors [24]. The technology allowed obtaining remarkable results in patients affected by spinal muscular atrophy (SMA), a childhood neuromuscular disease caused by the deletion or mutation of survival of motor neuron 1 (SMN1). The treatment involved the injection of antisense oligonucleotides (ASOs) [25,26] or viral vectors [27] in restoring SMN1 protein levels [28]. These results have opened doors to new treatments for fALS patients, and gene therapy clinical trials have been proposed.

Genetic interventions in ALS animal models, which targeted not only ALS-linked mutations, have been useful to understand the role of several proteins and pathways in the progression of the disease. Some studies have shown the importance of skeletal muscle in ALS progression, highlighting that motor neuron degeneration is not the only cause of alterations in this tissue. For instance, the modifications of trophic factors, such as glial cell-derived neurotrophic factor (GDNF), vascular endothelial growth factor (VEGF), and insulin-like growth factor 1 (IGF-1), in particular, improved ALS symptoms and survival. 

GDNF is a member of the TGF-β family, it was originally isolated in rat glial cells, and it promotes the differentiation and survival of dopaminergic neurons by increasing their dopamine uptake [29,30]. GDNF also has a neurotrophic function in motor neurons, where it enhances survival by preventing apoptosis and degeneration [31,32,33,34,35,36]. Because of this function, GDNF has been proposed as a therapeutic target for ALS [31,36,37]. Its expression in the skeletal muscle of 5–7-day-old SOD1^G93A^ mice slowed down the progression of the disease. The mice were injected with adenoviral vectors (AVR) into the hind limbs and the paraspinal muscles. In treated mice, the disease onset, the reduction of motor performance, and the motor neuron loss were delayed. Moreover, SOD1^G93A^-GDNF mice survived about two weeks longer than control mice [38]. The reason why the intramuscular injection of GDNF increased the lifespan is not clear, but it could be due to the effect on the NMJ, as GDNF increases the nerve sprouting capacity [38]. Alternatively, GDNF expression in skeletal muscle could promote motor neuron preservation in the spinal cord. Indeed, Acsadi and colleagues detected the protein both in the muscles and the spinal cord, probably because of the retrograde transport of motor neurons [38]. Consistent with these findings, the intramuscular transplantation with human mesenchymal stem cells engineered to secrete GDNF (hMSC-GDNF) protected SOD1^G93A^ rats from motor neuron loss and denervation, significantly delayed motor decline and increased the lifespan by about four weeks [39].

VEGF is another neuroprotective factor that may play a role in ALS. VEGF promotes angiogenesis and neuronal survival [40], as shown by the knockout of VEGF in wild-type mice that led to neurodegeneration and ALS-like symptoms [41]. Azzuoz and colleagues observed that a single injection of VEGF-expressing lentiviral vector (EIAV-VEGF) in different muscles of SOD1^G93A^ mice (gastrocnemius muscle, diaphragm, intercostal, facial, and tongue muscles) had positive effects on ALS symptoms [42]. To replicate potential clinical applications, two groups of mice received the EIAV-VEGF injection at two different time points: one before the onset (21 days of age) and the other at the onset of the disease. Both groups showed longer survival, better motor performances, and delays in motor neuron loss and motor weakness compared to controls [42].

The neurotrophin neuregulin 1 (NRG1) protects motor neurons from degeneration and is involved in the development and maintenance of axons [43,44,45] and NMJs, where it induces the clustering of acetylcholine receptors (AchRs) [46,47,48]. NRG1 exerts its functions by interacting with ErbB receptors; this interaction is impaired in ALS patients and animal models [49,50], which show low levels of expression of ErbB4 mRNA and protein in their skeletal muscle [51]. Similarly, the gastrocnemius muscle of SOD1^G93A^ mice has reduced levels of ErbB4 mRNA, which correlates with denervation [51]. The injection of an adeno-associated viral vector (AAV) expressing NRG1-I into the gastrocnemius muscle of SOD1^G93A^ mice significantly increased the muscle action potential and the collateral sprouting of axons [51]. In line with these results, expressing NRG1 in the skeletal muscles using the human desmin (hDesmin) promoter delayed the onset of ALS and improved the phenotype in SOD1^G93A^ mice. Indeed, NRG1 was shown to activate cell survival pathways in muscles and the spinal cord, protecting against denervation, neuroinflammation, and motor neuron loss. As the NMJs were preserved, treated mice had better neuromuscular and motor functions [52].

In ALS, NMJ degradation is the first event of denervation, and it occurs before motor neuron loss [14,53]. NRG1-ErbBs signaling and, hence, NMJ development and maintenance depend on the activation of muscle-specific receptor tyrosine kinase (MuSK) [54]. MuSK orchestrates the muscle-derived retrograde signal through the interaction with LRP4 and agrin, ensuring NMJ stability and maintenance while preventing disassembly [55,56,57,58]. MuSK overexpression in SOD1^G93A^ double transgenic mice delayed the onset of ALS, improved motor ability, and preserved the integrity of NMJs [59].

DOK-7 (docking protein 7) controls MuSK activation and response to agrin [60]; mutations in the DOK7 gene are responsible for the congenital myasthenic syndrome, which is characterized by impaired NMJ structure and functionality [61]. Since the administration of a recombinant AAV carrying the human DOK7 gene (AAV-D7) improved motor abilities and survival of animal models of DOK-7 myasthenia [62] and Emery–Dreifuss muscular dystrophy [63], Miyoshi and colleagues tried this therapeutic approach in ALS models: intravenous injection of AAV-D7 in SOD1^G93A^ mice at the onset of the disease significantly increased motor abilities, counteracted muscle atrophy, preserved NMJs, and extended the lifespan by more than ten days [64].

The microRNA miR206 and class II histone deacetylase 4 (HDAC4) control the denervation-reinnervation process [65]. The involvement of miR206 and HDAC4 in ALS progression has been proposed since their expression is altered in the skeletal muscles of patients and animal models [65,66,67,68]. HDAC4 is considered as a link between neuronal activity and muscle transcription because of its response to denervation [69]: HDAC4 expression levels in skeletal muscles increase during denervation, activating the muscle atrophy pathway driven by E3 ubiquitin-ligases MuRF1 and atrogin-1 transcription [69,70,71]. HDAC4 mRNA is upregulated in the muscle biopsies of ALS patients and correlates with disease severity, as the expression is higher in patients with a faster progression of the disease [68].

Pharmacological inhibition of class II HDACs in SOD1^G93A^ mice increased skeletal muscle electrical potential and improved motor abilities; however, it had no effect on survival and motor neuron loss [72]. On the other hand, the knockout of HDAC4 in skeletal muscle of SOD1^G93A^ mice worsened the ALS-like phenotype by speeding up the onset, the muscle force decline, and the NMJ loss, demonstrating that HDAC4 specifically has a protective role in ALS [73]. Indeed, HDAC4 induces the reinnervation pathway by activating the transcription of MuSK, miR206, and synaptic AchR through the indirect regulation of myogenin expression [69,70,71].

miR206 could be a prognostic marker for ALS, as high levels in serum correlate with a slower disease progression [74]. SOD1^G93A^ mice genetically deficient in miR206 showed worse ALS symptoms, shorter survival, increased muscle atrophy, and early NMJ loss, corroborating the important role of miR206 in the reinnervation process [65]. Williams and colleagues also observed that miR206 upregulation after denervation was higher in fast-twitch fibers than in slow-twitch fibers, probably because slow-twitch fibers express higher levels of miR206 in physiological conditions [65]. Interestingly, fast-twitch fibers are more vulnerable to denervation in ALS [75,76]; thus, the upregulation of miR206 could be a defense mechanism to protect them [65,77].

The protective role of miR206 in ALS could also be due to its role in satellite cell differentiation: miR206 promotes the differentiation of satellite cells into muscle cells by inhibiting PAX7 expression; PAX7 inhibition is further enhanced by a positive feedback loop in which the muscular differentiation factors MyoD, myogenin, and MEF2 induce miR206 transcription [78,79,80].

In adult muscles, MyoD is predominantly expressed in fast-twitch fibers, while myogenin in slow-twitch fibers [81,82,83], where it stimulates oxidative metabolism [84,85]. Given their differential role and expression patterns, Park and colleagues hypothesized that the postnatal expression of MyoD and myogenin in muscles could affect ALS progression in opposite ways [86]. Indeed, MyoD overexpression in 30-day-old SOD1^G93A^ mice via intramuscular injection with an AVV vector led to a more aggressive phenotype, shorter survival, earlier decline of motor performances, and premature loss of motor neurons and NMJs [86]. On the other hand, myogenin overexpression in skeletal muscle improved motor functions and preserved innervation and motor neurons; however, there were no changes in the lifespan [86].

The results from MyoD and myogenin overexpression and miR206 deletion led to the hypothesis that fibers in ALS switch from the fast to the slow type to preserve integrity and functionality of motor units and skeletal muscle [87]. This switch may reflect the metabolic shift towards oxidative metabolism occurring in ALS patients and animal models [88,89,90]. Since mitochondria regulate metabolism plasticity, they may also play a crucial role in the ALS fiber switch [90]. In ALS mouse models, mitochondrial dysfunctions and energy production impairments occur before the onset of the disease in skeletal muscle and at the onset of the disease in the spinal cord [90,91,92]. In addition, it has been shown that mitochondrial dysfunctions start before muscle differentiation, as they can be detected in the satellite cells of SOD1^G93A^ mice [90].

Studies have shown that genetic interventions that enhance mitochondrial performance can improve muscle functionalities and quality of life. Peroxisome proliferator-activated receptor-gamma coactivator-1a (PGC-1a) is a major regulator of mitochondrial biogenesis and activity [93,94]. In SOD1^G37R^ mice, the overexpression of PGC-1a in skeletal muscle improved muscle performance, locomotor activity, resistance to fatigue, and muscle atrophy; it also increased the mitochondrial area and the oxygen consumption in skeletal muscle [86,95]. Consistent with these results, the overexpression of uncoupling protein 1 (UCP1) (a mitochondrial protein that mediates non-shivering thermogenesis by uncoupling the mitochondrial electron transport chain) shortened the lifespan of SOD1^G86R^ mice and increased disease duration and progression [96]. UCP1 overexpression in skeletal muscle of wild-type mice was sufficient to trigger NMJ dismantlement and distal motor neuron degeneration [96], highlighting the importance of mitochondria for the integrity and functionality of skeletal muscle. Together, these results show that enhancing mitochondrial functions and regulating the energy homeostasis of muscles can delay disease progression and improve the quality of life. However, it should be noted that improving mitochondrial performance in mouse models of ALS does not always produce an increase in survival, as described by some works that through genetic or pharmacological approaches improve mitochondrial proliferation [97,98,99].

IGF-1 regulates skeletal muscle physiology as well as mitochondrial dynamics and turnover [100,101,102]; it prevents inflammation, controls protein synthesis and degradation, and promotes satellite cell proliferation and neuronal survival [101,102,103,104]. The expression of IGF-1 in skeletal muscle of ALS models gave the most remarkable results on disease progression and survival, delaying the death of SOD1^G93A^ mice by about one month [105]. In these mice, the regeneration pathways through calcineurin and CDK5 were induced, while apoptotic and ubiquitin pathways were inhibited, protecting muscles against atrophy and denervation and preserving NMJs and motor neurons [105,106]. Interestingly, high concentrations of IGF-1 in patients’ serum correlate with a better prognosis but not with a lower risk of ALS, suggesting that IGF-1 plays a role in the survival of ALS patients [107].

All preclinical genetic interventions on skeletal muscle highlighted the importance of this tissue in ALS progression as modifying the expression of genes involved in skeletal muscle physiology, metabolism, and functions had a strong impact (either positive or negative) on the disease. Therefore, muscle-directed gene therapy could become a therapeutic approach for ALS.

Table 1 summarizes the genetic interventions on ALS skeletal muscle.

## 3. Pharmacological and Nutritional Interventions

To date, pharmacological interventions to counteract ALS neuromuscular degeneration target mainly neurons. Drugs targeting skeletal muscle have been tested only recently; they could improve energetic metabolism and allow sprouting and formation of new synapses. Enhancing muscle and, hence, respiratory functions should be a priority in ALS care, also because it can improve the patient’s quality of life.

In this section, we have classified the pharmacological treatments based on their action on pathological events; however, considering their tight interdependence, there is a fine line between the different skeletal muscle alterations.

### 3.1. Pharmacological Interventions That Target Hypermetabolism

As muscle activity is the major contributor to the whole-body energy metabolism, muscles are likely to play a crucial role in ALS hypermetabolism. Defects in muscular ATP production and altered substrate utilization have been reported in patients and animal models [53,108]. In ALS, fiber transition from fast fatigable to fast intermediate and fast fatigue-resistant occurs before any measurable locomotor defects [88,89,90]. As shown in mSOD1 ALS mice, this transition results in the switch from glycolysis (i.e., the use of glucose as the main energy source) to ß-oxidation (i.e., the use of fat as the main energy source) [88,89,90]. Consistent with these findings, a study found that many ALS patients experience a sharp decrease in BMI and weight due to lipid consumption almost ten years before the onset of the disease and the appearance of motor symptoms [20].

Different metabolic therapies have been tested on animal models and patients to sustain their fatty acid consumption.

Carnitine is a fundamental source of acetyl groups. As it acts by transporting long-chain fatty acids into the mitochondrial matrix, its bioavailability is directly related to the rate of ß-oxidation [109]. 95% of carnitine resides in skeletal muscle since this tissue largely depends on fatty acids as an energy source [110].

Carnitine affects muscle remodeling by preventing atrophy and activating the oxidative stress response [111]. In presymptomatic SOD1^G93A^ mice, oral administration of L-carnitine extended the lifespan while delaying motor impairment and disease onset [112]. Similar results were also obtained with subcutaneous injection at the onset of the disease [112]. The co-administration of acetyl-L-carnitine and riluzole in a small group of ALS patients was well tolerated and resulted in a better ALSFRS score (ALS Functional Rating Scale) as compared to riluzole alone [113] (EudraCT number: 2004-004158-23). Despite these encouraging results, a larger Phase III trial to validate the effectiveness of the treatment is still missing.

A high-fat diet (HFD) in SOD1^G93A^ mice extended the mean survival by about 20% [114], and a high-energy diet based on medium-chain fatty acids and beta-hydroxybutyrate reduced locomotor defects in a Drosophila model of ALS [115]. Interestingly, a study involving more than two hundred patients fed with HFD showed that this diet significantly extended the survival of fast-progressing patients [116].

Creatine has been recently proposed in preclinical and clinical studies to compensate for the progressive depletion of energy reserves in ALS. Creatine is an amino acid endogenously synthesized or found in food, which is mainly absorbed by skeletal muscle. Creatine is phosphorylated by creatine kinase (CK) to phospho-creatine (PK) that is used as a source of energy during rapid and intense muscle contractions [117,118,119,120]. Since it helps improve muscle performance, creatine is often taken by athletes as a dietary supplement [121]. Initially, preclinical studies provided encouraging data as creatin was shown to delay the impairment of locomotor functions and extend lifespan [122,123,124]; however, a later study did not confirm these results [125]. Similarly, randomized controlled human trials evaluating the efficacy of creatine monohydrate, administered alone or in combination with other drugs (NCT00005766, NCT00005674, NCT00355576, NCT00070993, NCT00069186, NCT01257581) [126,127,128], showed that this compound did not improve disease progression or survival in ALS patients [129]. However, high CK levels have been correlated with a slower progression of the disease in ALS patients and mouse models [130], suggesting that providing supplements to the muscles could partially compensate for the catabolic effects of ALS hypermetabolism.

Inhibiting the β-oxidation of fatty acids to induce glycolysis has been another strategy to counter hypermetabolism in ALS. A recent study of SOD1^G93A^ mice showed that starting the chronic administration of Ranolazine, an inhibitor of β-oxidation, at the onset of the disease slowed down the muscle strength loss and improved the motor functions, but not the lifespan. We correlated the administration of Ranolazine with improvements in energy metabolism, as the drug reduced the whole-body energy expenditure of SOD1^G93A^ mice by increasing the levels of ATP in muscles [90].

Consistent with these results, dichloroacetate (DCA), a pyruvate dehydrogenase kinase 4 (PDK4) inhibitor that switches muscle metabolism from β-oxidation to glycolysis, improved muscle strength, maintained NMJ integrity and reduced the expression of denervation markers in SOD1^G86R^ mice [89]. Its administration in presymptomatic SOD1^G93A^ mice delayed the onset of the disease, enhanced motor performance and increased the lifespan by improving the mitochondrial redox status [131]. However, although its use in certain chemotherapies, DCA has severe side effects, hepatotoxicity in particular [132], making it not suitable for long-term treatment of neurodegenerative diseases.

Together, these results indicate that hypermetabolism, and thus skeletal muscle, can be good drug targets for ALS.

### 3.2. Pharmacological Interventions to Increase Muscle Mass

Several pathological phenomena, including defects in the proliferation and differentiation processes, lead to skeletal muscle mass loss in ALS [53,133]. Different pharmacological approaches have been tried to counter mass loss, such as the treatment with anabolic androgenic steroids (AAS).

AAS are synthetic derivatives of the testosterone hormone that increase muscle mass (and, for this reason, are also often used illegally by athletes to enhance performances). Subcutaneous administration of dihydrotestosterone crystals (an AAS) in early-symptomatic SOD1^G93A^-induced weight gain, reduced muscle atrophy, increased grip strength, and extended lifespan [134]. Interestingly, dihydrotestosterone treatment increased muscular expression of IGF-1, which protects mitochondria of murine and cellular models of ALS by increasing mitophagy and upregulating the expression of anti-apoptotic proteins [135] (see above). Similarly, the chronic administration of the AAS nandrolone in presymptomatic SOD1^G93A^ mice maintained the mass of the diaphragm muscle, despite mild side effects on muscle fiber innervation [136]. However, an earlier study on the same mouse model showed that the administration of nandrolone significantly increased the expression of TGFβ1a in muscles [137], suggesting that it could even worsen the disease [138].

Myostatin inhibits myogenesis and muscle growth by reducing fiber number and size [139,140,141,142]. Its overexpression led to weight loss, muscle atrophy, and sarcopenia [143], while its downregulation to muscle hypertrophy and a hypermuscular phenotype [144,145]. Because of its impact on muscle mass, several drugs that inhibit the myostatin signaling pathway have been evaluated in preclinical and clinical studies to treat a variety of muscle-wasting diseases [146]. Myostatin levels are significantly higher in ALS patients than in healthy individuals and they are positively correlated with the rate of muscle degeneration [147]. In two rodent models of ALS, neutralizing antibodies against myostatin decreased the weight loss and increased the mass and strength of muscles at the onset of the disease and during the early-stages [142]. However, they did not delay the disease onset nor increased survival [142]. Similar results were obtained in SOD1^G93A^ mice using an Fc chimera of the activin receptor type IIB, an endogenous signaling receptor for myostatin [148].

Although obtained only in preclinical studies, these data suggest that targeting myostatin signaling pathways may have a therapeutic effect in ALS. Supporting this hypothesis, anti-myostatin antibodies improved muscle performance in a mouse model of Duchenne muscular dystrophy and prevented skeletal muscle alterations in a Huntington’s disease mouse model [149]; moreover, the administration of follistatin, a natural antagonist of myostatin, improved the severity of SMA in mice [150].

### 3.3. Pharmacological Interventions to Preserve NMJs and Reduce Atrophy

NMJ dismantling and muscle atrophy are early events in ALS and precede denervation, supporting the idea that skeletal muscle plays a key role in the disease. The maintenance of NMJs and the inhibition of atrophic processes have been considered as main targets for pharmacological interventions.

ALS patients’ muscles aberrantly express the neurite growth inhibitor Nogo-A, and its levels correlate with the severity of symptoms [151,152]. In ALS, Nogo-A causes retrograde axonal degeneration by destabilizing the NMJs. The overexpression of Nogo-A in murine healthy muscle fibers induced the detachment of NMJs, while its genetic ablation in SOD1^G86R^ mice reduced denervation and increased the lifespan [151]. Despite these encouraging results, the administration of the anti-Nogo-A monoclonal antibody ozanezumab was ineffective in a phase II trial (NCT01753076) [153].

As discussed above, activating MuSK has been another approach to prevent or delay NMJs dismantling. Although ALS patients and animal models do not have alterations in the MuSK pathway, two preclinical studies have stimulated MuSK with an agonist antibody. These studies obtained conflicting results despite using a similar approach (same antibody and same mouse model). In the first study, the administration of the MuSK agonist preserved NMJs, delayed denervation and increased survival in SOD1^G93A^ mice [154]. The second study did not report any improvements in diaphragm functionality or lifespan, despite the retention of NMJs in the diaphragm [155]. Therefore, further investigations are needed to understand the effectiveness of this therapeutic strategy.

Muscle acetylcholine receptors (AchRs) have also been proposed as therapeutic targets in ALS. In a recent clinical study, the administration of palmitoylethanolamide (PEA), an endocannabinoid that reduces the desensitization of AchRs currents following repeated stimulation, improved pulmonary functions and delayed the decrease of forced vital capacity (FVC) (NCT02645461) [156]. The authors also reported that PEA strongly upregulated the expression of the α1 AChR subunit and that it maintained NMJ functionality by reducing the rundown of ε-AChRs currents. Interestingly, another study showed that riluzole blocked muscle AchRs with greater specificity for γ-AChRs than ε-AChRs; however, the resulting biological consequences were not clarified [157].

### 3.4. Other Pharmacological Interventions

Fast skeletal muscle troponin activators (FSTA), which selectively activate the troponin complex and increase its sensitivity to calcium, have been studied as potential treatments for ALS [158,159]. Troponin is a protein complex that modulates muscle contractility and increases muscular strength and power, slowing down the onset of fatigue, particularly in respiratory muscles. The FSTA tirasemtiv gave good results in both preclinical [159] and early clinical studies (NCT01486849; NCT01089010; NCT02936635; NCT01709149; NCT01378676), maintaining muscle strength and delaying the onset and the level of muscle fatigue. However, in the phase III VITALITY-ALS trial (NCT02496767) involving 81 centers in the United States and Europe, tirasemtiv did not impact the decline of slow vital capacity (SVC), nor secondary outcomes, such as the ALSFRS-R score, the first use of mechanical ventilatory assistance, and death [160]. The fact that many patients did not tolerate the treatment and left the trial may have contributed to its disappointing results [160]. Following these data, reldesemtiv, a next-generation FSTA compound, was synthesized. Its functions are similar to tirasemtiv, but it has different chemical characteristics. A double-blind, randomized, placebo-controlled, variable dosage trial (NCT03160898) tested the safety of reldesemtiv and demonstrated that the drug was well tolerated by patients and that there was a trend towards improvement in primary and secondary outcomes, though it was not statistically significant [161]. The therapeutic potential of reldesemtiv is currently studied for the treatment of other diseases associated with muscle dysfunction and weakness, such as SMA, chronic obstructive pulmonary disease (COPD), and in elderly subjects with reduced mobility (NCT02644668sma- NCT03065959-copd).

The FSTA levosimendan has been recently tested. This compound showed positive effects in a phase II clinical trial [162] (NCT02487407); however, phase III clinical trials did not confirm the data, as the oral administration of levosimendan did not improve the respiratory function nor the general functions of ALS patients (NCT03505021; NCT03948178). In light of these discouraging results, this pharmacological approach has now been abandoned.

Recently, different studies have shown that CTGF/CCN2, a member of the CCN family of extracellular matrix-associated heparin-binding proteins, is upregulated in skeletal muscle and spinal cord of ALS patients [163,164]. CTGF/CCN2 plays a crucial role in tissue fibrosis, as it affects angiogenesis, migration, proliferation, and cell adhesion [165]. Treating SOD1^G93A^ mice with a monoclonal neutralizing antibody against CTGF/CCN2 (FG-3019) improved locomotor performance and reduced muscular fibrosis and atrophy [163]. Preclinical studies using this treatment have provided encouraging results also for other pathologies associated with fibrosis, including skeletal muscle dystrophies [166].

Another therapeutic approach using aminophylline, which is supposed to mainly act on smooth muscle, was tested. Aminophylline is a soluble derivative of theophylline, a compound that relaxes smooth muscles and relieves bronchial spasm. Theophylline functions as a phosphodiesterase inhibitor, an adenosine receptor blocker, and a histone deacetylase activator [167]; it is widely used for the treatment of asthma, bronchospasm, and COPD [168,169,170]. Two studies showed that theophylline improved the strength and endurance of peripheral and respiratory muscles [171,172]. In a double-blind, randomized crossover trial, 25 ALS patients with a disease duration of fewer than five years received intravenous aminophylline; the treatment improved the endurance of respiratory muscles and increased handgrip strength [173]. Despite these promising results, the therapeutic potential of aminophylline in ALS has not been further investigated.

Finally, the activation of P2X7, a purinergic receptor abundantly expressed in skeletal muscle, with the specific agonist 2′(3′)-O-(4-benzoylbenzoyl) adenosine 5′-triphosphate improved muscle metabolism and preserved NMJ morphology in presymptomatic SOD1^G93A^ mice. Interestingly, P2X7 is a key regulator of myofiber differentiation and regeneration [174].

Together, these data highlight the possibility to target skeletal muscle with a wide range of drug classes.

Preclinical and clinical studies are summarized in Table 2 and Table 3, respectively.

## 4. Physical Exercise as a Therapeutic Approach

Adult skeletal muscle is a highly plastic tissue that adapts in response to external stimuli [175]. For instance, variations in nutrient intake, aerobic, anaerobic conditions, and hormonal responses determine the structure of skeletal muscle. In addition, muscles adapt to physical training with structural and physiological changes, leading to positive health impacts [176].

The benefits of physical activity on motor neuron loss and sarcopenia are widely recognized [177]. As exercise may improve several chronic conditions, it could be considered as a therapeutic intervention to slow down muscle degeneration and preserve NMJ integrity in ALS.

Besides mental and other biological improvements, physical activity preserves specific mechanisms altered in the progression of ALS. For instance, exercise triggers pathways that improve skeletal muscle metabolism, enhance muscle glucose utilization and induce myofiber regeneration by activating satellite cells [178]; moreover, regular physical activity, regardless of the type, strengthens antioxidative defenses [179], and endurance training increases mitochondrial biogenesis in skeletal muscle [180] and neurogenesis [181,182]. However, preclinical studies have so far provided contradictory results, suggesting that the outcomes depend on the type and intensity of physical activity.

The adaptive response to exercise is a hormetic response that follows a biphasic curve, where low levels of stimuli elicit beneficial effects, whereas chronic and/or high levels of the same stimuli lead to negative or even toxic effects [183]. According to the exercise-induced hormesis theory, regular moderate-intensity training counteracts free radicals-induced cell injuries and inflammation processes [184,185,186], improves cardiovascular functions [187,188,189] and protects from different senescence-related processes [190,191], such as mitochondrial alterations in skeletal muscle [192,193,194]. On the other hand, continuous and high-intensity training triggers opposite effects, speeding up aging processes and increasing oxidative stress [195,196].

Models of ALS have shown the effects of exercise-induced hormesis. For instance, the phenotype of SOD1^G93A^ mice that exercised on a motorized treadmill improved only with moderate exercise intensity, whereas high exercise intensity speeded up the decline of motor performance and did not preserve the density of motor neurons in the lumbar spine ventral horn [197]. In line with these observations, SOD1^G93A^ mice exercising on a running wheel at moderate intensity showed a modest improvement in lifespan and locomotor performances [198,199,200]; on the other hand, intense, forced treadmill exercise worsened their phenotype [201].

Kirkinezos et al. and Veldink et al. showed that the effects of exercise on ALS phenotype depended on gender; however, they reported opposite conclusions. In both studies, SOD1^G93A^ mice run daily on a treadmill for 30 and 45 min, respectively, at moderate intensity [202,203]. Kirkinezos et al. concluded that physical activity benefitted only in male mice [202], while Veldink et al. observed a positive neuroprotective effect only in females [203]. The authors proposed a different role of sex hormones to explain the gender-specific responses. However, in both studies, the mice were forced to run using an electric shock; this method has likely introduced biases in the results because of the stress that is induced in the animals. Garbugino et al. described the effects of voluntary exercise in low-copy SOD1^G93A^ mice running on a home-cage running-wheel system [204]. The authors concluded that male mice were worse affected by prolonged and repeated exercise than females, showing shorter survival, increased body weight loss, and poorer prognosis. Their results were in line with the theory of exercise-induced hormesis [204].

The type of exercise may also affect the ALS phenotype. While running exercises have provided controversial results, the benefits of swimming-based exercises have been clear, as Deforges et al. showed that swimming extended the lifespan of SOD1^G93A^ mice by about 25 days [205]. The authors hypothesized that swimming and running involve different motor units: while swimming is characterized by high-frequency and large-amplitude movements mainly recruiting the fast motor units, mostly running triggers the slow motor units [205]. These results were confirmed by other papers describing the beneficial effects of swimming on the phenotype of ALS mouse models [206,207,208,209].

Motor neuron fast-twitch fibers, which are preferentially stimulated when swimming, degenerate first and are more compromised in ALS [210], probably because they use the glycolytic pathway as their main energy source, a pathway heavily impaired in ALS [88,89,90]) (see above). Swimming improved glucose metabolism in SOD1^G93A^ mice more efficiently than running [209]; specifically, it induced the expression of glucose transporter GLUT4 and of glyceraldehyde-3-phosphate dehydrogenase (GAPDH), the key enzyme of the glycolytic pathway, countering the glucose intolerance of SOD1^G93A^ mice [88,90].

Exercise, and swimming, in particular, reduced the deregulation of the BDNF/TrkB pathway in SOD1^G93A^ mice [211]. BDNF, a neurotrophin secreted following muscle contraction, can be either a neuroprotector or a neurotoxin by probably acting in a paracrine way and increasing glutamate excitotoxicity through the activation of TrkB receptors [212,213]. In SOD1^G93A^ mice, the neuronal hyperexcitability and the following muscle contractions induced the over-secretion of BDNF that likely contributed to neurodegeneration by enhancing glutamate toxicity [211]. A recent paper showed that preserving the BDNF/TrkB pathway through a specific swimming-based training improved the phenotype of ALS mice [206]. Consistent with these findings, 70 to 115-day-old SOD1^G93A^ mice trained by swimming (performed in an adjustable-flow swimming pool) and running (performed on a treadmill at moderate intensity) showed a decrease in muscular BDNF concentration; in particular, muscular BDNF reached physiological levels in the mice that underwent the swimming-based training [206].

Overall, preclinical studies have shown that mild-to-moderate aerobic training improves the phenotype of ALS animal models. A recent meta-analysis assessed the impact of exercise on ALS patients by comparing 94 patients that underwent therapeutic exercise with 159 patients treated with conventional therapy [214]. The authors concluded that exercise could positively affect the rate of weakening of physical functions; however, these results should be interpreted with caution due to the limited number of studies and the different protocols used [214].

Although the therapeutic use of physical activity in ALS patients is still debated, a recent randomized controlled study on 22 patients showed that repetitive twitches induced by local magnetic stimulation hampered muscle atrophy, increased local muscular strength, and slowed down the metabolic shift towards ß-oxidation [215]. The molecular analysis of muscular biopsies showed that the magnetic stimulation counteracted muscle atrophy and proteolysis by increasing the efficacy of nicotinic ACh receptors [215].

ALS patients have defects in the energetic metabolism of skeletal muscle and alterations in energy expenditure, which indicate a poor prognosis [16]. The hypothesis that lifetime physical activity is a risk factor for developing ALS is under debate [216] and has raised doubts about the use of physical activity, which increases the body’s energy requirements, as a therapy. These doubts are also justified by clinical evidence showing that the oxidative capacity of skeletal muscle is impaired when ALS patients undergo intense physical exercise [217,218,219]. A study also described a mild mitochondrial dysfunction at the onset of the disease that could be detected only during exercise [218].

The analysis of the oxidative capacity of skeletal muscle in exercised patients has provided heterogeneous results depending on their exercise capacity and clinical profile [218]. Overall, the data have highlighted the need to adapt the type and intensity of physical activity to each patient. For instance, Ferri and colleagues showed that a training program combining moderate-intensity aerobic and strength training improved the patients’ aerobic capacity and physical function when tailored to their individual needs [220].

Consistent with the effectiveness of training programs that combine strength and endurance exercises, Lunetta et al. showed that patients doing strength exercises for the upper and lower limbs and exercises on a cycle ergometer at a moderate intensity improved their ALSFRS-R score; however, the training did not extend their survival [221]. These results were confirmed by a pilot randomized study from Merico et al. that showed how a combined exercise program improved the patients’ functional status measured with the functional independence measure (FIM) [222]. Resistance and strength exercises alone were also well tolerated by ALS patients and generally improved the patient’s quality of life [223]. Indeed, resistance training protocols at moderate intensity improved the scores of the ALSFRS-R test [224,225] and of the 30-second sit-to-stand test [226]. Endurance training performed with moderate aerobic exercises has also been shown to increase the ALSFRS-R score in certain cases [227,228].

In conclusion, physical activity as a therapy option has given interesting results, especially regarding the patient’s quality of life. However, the type of exercise should be tailored to each patient’s needs, and the intensity should always be moderate.

## 5. Concluding Remarks

Skeletal muscle has been long neglected in ALS, but recent data have highlighted its role in the etiopathogenesis of the disease.

In this review, we have discussed the preclinical and clinical studies that have targeted skeletal muscle to treat ALS. Since all of them emphasized the pivotal role of this tissue in ALS progression, skeletal muscle should be considered an optimal target site for therapeutic intervention.

In our opinion, although the results obtained so far have not introduced substantial innovations in clinical practice, they allow us to draw important conclusions:

Skeletal muscle is the main determinant of the whole-body energy expenditure, and interventions that improve its metabolism bring benefits to the entire organism.

Given that the functions of muscles and motor neurons are tightly intertwined, therapeutic interventions targeting skeletal muscle can counter the dying back process and, ultimately, protect motor neurons.

Both the physiology and the accessibility of muscle tissue make it a good therapeutic target that is worth considering at least to improve the patients’ quality of life.

## Figures and Tables

**Table 1 cells-10-00525-t001:** Genetic intervenctions.

Gene	Function	Expression Type in Muscle	Model	Effect	Survival	References
Glial cell-derived neurotrophic factor (GDNF)	Trophic effect on motor neurons	Expression by muscle injection (AVR) at not symptomatic stage (age: 5–7days)	SOD1^G93A^ mice	∧ Motor performance; ∨ motor neuron loss	YES	[38]
		Engineered human mesenchymal stem cells (hMSC-GDNF)	SOD1^G93A^ rats	∧ Motor performance; ∨ motor neuron loss; ∨ denervation	YES	[39]
Vascular endothelial growth factor (VEGF)	Angiogenesis and neuroprotection	Expression by muscle injection (EIAV) at not symptomatic stage (age: 21 days)	SOD1^G93A^ mice	∧ Motor performances; ∨ motor weakness; delayed onset	YES	[42]
		Expression by muscle injection (EIAV) at the onset (age: 90 days)	SOD1^G93A^ mice	∧ Motor performances; ∨ motor weakness; ∨ motor neuron loss	YES	[42]
Insulin-like growth factor 1 (IGF-1)	Anabolism of muscle and nerve tissues, myogenesis and neuronal survival	Transgenic mice, muscle restricted expression	SOD1^G93A^ mice	∧ Muscle regeneration; ∧ preservation NMJ; ∨ Muscle atrophy; ∨ MN loss; ∨ apoptotic and ubiquitin pathways	YES	[105,106]
MicroRNA-206 (miR-206)	Myogenesis, NMJ formation, stabilization and repair	Transgenic mice, muscle restricted deletion	SOD1^G93A^ mice	∧ Muscle atrophy; ∧ NMJ loss; ∧ disease progression; ∨ disease duration	NO	[65]
Uncoupling protein1 (UCP1)	Thermogenesis by uncoupling mitochondrial electron transport from ATP synthesis	Transgenic mice, muscle restricted overexpression	SOD1^G86R^ mice	∧ Disease progression; ∨ disease duration	NO	[96]
Muscle-specific kinase (MuSK)	Formation and maintenance of NMJ	Transgenic mice, muscle restricted overexpression	SOD1^G93A^ mice	∧ Motor performances; ∨ NMJ denervation; delayed onset	NO	[59]
Peroxisome proliferator-activated receptor-gamma coactivator- 1a (PGC-1a)	Cellular energy metabolism, mitochondrial biogenesis and angiogenesis	Transgenic mice, muscle restricted overexpression	SOD1^G937R^ mice	∧ Mitochondrial biogenesis; ∧mitochondria area; ∧ resistance to fatigue; ∧ Locomotor activity; ∧ Mitochondrial oxygen consumption in skeletal muscle; ∨ muscle atrophy	NO	[95]
			SOD1^G93A^ mice	∧ Muscle fiber oxidation; ∨ motor function	NO	[86]
MyoD	Muscle development and differentiation	Expression by muscle injection (AV) in adult mice (age: 30 days)	SOD1^G93A^ mice	∧ Weight loss; ∧ Motor neuron loss; ∨ motor performances; ∨ NMJ innervation; ∨ muscle fiber oxidation	NO	[86]
Myogenin	Muscle development and differentiation increases oxidative metabolism of muscle	Expression by muscle injection (AV) in adult mice (age: 30 days)	SOD1^G93A^ mice	∧ Motor performances; ∧ NMJ innerve tion; ∧ Muscle fiber oxidation; ∨ motor neuron loss	NO	[86]
DOK-7	Neuromuscular synapsis formation by regulation of Musk activity	Expression by intravenous injection (AAV) at the onset (age: 90 days)	SOD1^G93A^ mice	∧ NMJ innervation (diaphragm); ∧ Motor activity; ∨ muscle atrophy	YES	[64]
Histone deacetylase 4 (HDAC4)	Skeletal muscle response to denervation	Transgenic mice, muscle restricted deletion	SOD1^G93A^ mice	∧ Muscle atrophy; ∧ weight loss; ∨ muscle force; ∨ NMJ innervation; precocious disease onset.	NO	[72]
Neuregulin 1 (NRG1)	Axonal and neuromuscular development and maintenance	Overexpression by intramuscular injection (AAV) at the onset (age: 8 weeks)	SOD1^G93A^ mice	∧ Axons collateral sprouting and NMJ; ∧ Compound muscle action potential	N/A	[51]
		Overexpression by intravenous injection (AAV under hDesmin promoter) in adult mice (age: 6 weeks)	SOD1^G93A^ mice	∧ Neuromuscular functions; ∧ NMJ innervation; ∧ cell survival pathway activation; ∧ locomotor ability; ∨ Motoneuron loss; ∨ neuroinflammation; ∨ oxidative stress in skeletal muscle; delayed onset	N/A	[52]

**Table 2 cells-10-00525-t002:** Preclinical pharmacological interventions.

	Drugs	Function	Model	Effects	Survival	References
**Metabolic modulation**	L-Carnitine	Cofactor for the beta-oxidation of long-chain fatty acids	SOD1^G93A^ mice	Delayed deterioration of motor activity	YES	[112]
	Dichloroacetate	Improves glycolysis	SOD1^G86R^ mice	∧ Maintenance of NMJs; ∧ Muscle strength; ∨ denervation markers	YES	[89,131]
	Ranolazine	Inhibition of beta-oxidation	SOD1^G93A^ mice	∧ Motor functions; ∧ Muscle ATP; ∧ energy metabolism	NO	[90]
**Modulation of muscle mass growth**	Anti-Myostatin	Endogenous inhibitor of myogenesis	SOD1^G93A^ mice SOD1^G93A^ rats	∧ Muscle mass strength; ∨ weight loss	NO	[142]
	ActRIIB.mFc	Endogenous signaling receptor for myostatin	SOD1^G93A^ mice	∧ Body weight; ∧ grip strength; ∧ Muscle size	NO	[148]
	Dihydrotestosterone	Activator of anabolic functions	SOD1^G93A^ mice	∧ Weight loss; ∧ grip strength	YES	[133]
	Nandrolone	Activator of anabolic functions	SOD1^G93A^ mice	∧ Diaphragm muscle mass	NO	[136]
**NMJ preservation and atrophy reduction**	Anti-Musk	Development and stability of NMJs	SOD1^G93A^ mice	∧ Muscle mass; ∧ strength; ∨ muscle; ∨ denervation	YES	[154]
				∧ innervation of the neuromuscular junction; ∨ diaphragm function, motor neurons	NO	[155]
**Other interventions**	Tirasemvit (CK-357)	Fatigue resistance of the muscle	SOD1^G93A^ mice	∧ Submaximal isometric force; ∧ forelimb grip strength; ∧ grid hang time; ∧ rotarod performance; ∧ diaphragm force	NO	[159]
	FG-3019	Development and stability of NMJs	SOD1^G93A^ mice	∧ Locomotor; ∧ performance; ∨ muscular fibrosis; ∨ atrophy	NO	[160]
	2′(3′)-O-(4-Benzoylbezoyl) Adenosine5′-triphosphate (BzATP)	P2X7 agonist	SOD1^G93A^ mice	∧ Muscle metabolism; ∧ NMJs morphology	NO	[174]

**Table 3 cells-10-00525-t003:** Clinical pharmacological interventions.

	Drugs	Function	Phase	Clinical Trial	References
	Acetyl L-carnitine	Cofactor for the beta-oxidation of long-chain fatty acids	II	EudraCT Number: 2004-004158-23	[113]
**Metabolic** **modulation**	Creatine	Facilitates recycling of adenosine triphosphate (ATP), the energy currency of the cell, primarily in muscle and brain tissue.	II II II II III	NCT00005766 NCT00005674 NCT00355576 NCT00070993 NCT00069186	[126,127,128]
			II	NCT01257581	
**NMJ** **preservation and atrophy reduction**	Ozanezumab	Humanized monoclonal antibody against Nogo-A	II	NCT01753076	[153]
**Other** **interventions**			II	NCT01486849	
			II	NCT01089010	
	Tirasemvit (CK-357)	Fast skeletal muscle troponin activators (FSTA)	III II	NCT02936635 NCT01709149	[163]
			II	NCT01378676	
			III	NCT02496767	
	Reldesemtiv (CK-2127107)	Protein complex that modulates muscle contractility and increases the strength and power of the muscular system	II	NCT03160898	[161]
			II	NCT02487407	
	Levosimendan	Increases the functionality of the musculoskeletal system	III	NCT03505021	[162]
			III	NCT03948178	
	Palmitoylethanolamine	Analgesic and anti-inflammatory	N/A	NCT02645461	[158]
	(PEA)				
	Aminophylline	Adenosine receptor antagonist	N/A	N/A	[173]

## Data Availability

No new data were created or analyzed in this study. Data sharing is not applicable to this article.

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
