# Peer review of "Skeletal Muscle in ALS: An Unappreciated Therapeutic Opportunity?"

_cells, 2021, doi:10.3390/cells10030525_

Round 1
Reviewer 1 Report
The review paper titled” Skeletal muscle in ALS: an unappreciated therapeutic opportunity?”
Focus the attention on skeletal muscle alteration beyond motor neuron disease in ALS. It is very intersting that skeletal muscle alterations, described in the early stages of the disease, seem to be mainly involved in the “dying back” phenomenon of motor neurons and metabolic dysfunctions.
Nevertheless, some references need to be added and discuseed.
In line 54-61 there is is lack in references.
To implement the reviwe the authors need to cite and discuss the following literature
-Natale G, Lenzi P, Lazzeri G, Falleni A, Biagioni F, Ryskalin L, Fornai F. Compartment-dependent mitochondrial alterations in experimental ALS, the effects of mitophagy and mitochondriogenesis. Front Cell Neurosci. 2015 Nov 6;9:434. doi: 10.3389/fncel.2015.00434.
-Parone et al. (Parone, P. A., Da Cruz, S., Han, J. S., McAlonis-Downes, M., Vetto, A. P., Lee, S.K., et al. (2013). Enhancing mitochondrial calcium buffering capacity reduces aggregation of misfolded SOD1 and motor neuron cell death without extending survival in mouse models of inherited amyotrophic lateral sclerosis. J. Neurosci. 33, 4657–4671. doi: 10.1523/JNEUROSCI.1119-12.2013).
Ruffoli et al., Ultrastructural studies of ALS mitochondria connect altered function and permeability with defects of mitophagy and mitochondriogenesis. Front Cell Neurosci. 2015 Sep 1;9:341. doi: 10.3389/fncel.2015.00341. eCollection 2015.
Author Response
Thank you for appreciating our manuscript. The appropriate references have now been inserted in the introduction section.
We implemented the review citing and discussing the suggested literature in the paragraph "Genetic interventions”:
However, it should be noted that improving mitochondrial performance in mouse models of ALS does not always produce an increase in survival, as described by some works that through genetic or pharmacological approaches improve mitochondrial proliferation [98–100].Reviewer 2 Report
The authors wrote a review on the role of skeletal muscles in ALS and its potential as a therapeutic target. The review focusses on studies that target skeletal muscles. The review is quite comprehensive and includes a wide range of approaches such as genetic and pharmacological interventions, lifestyle and dietary approaches. I believe that this review would be very useful to researchers working in the field as the topic is timely and given the recent advances in the field, the other reviews on this topic might be slightly outdated already.
The work is well structured and clear. I have only a few minors:
1) The first part of the introduction lacks appropriate references. Paragraph 1-3 (line 30-45) and 5 (line 54-60) have no references at present and I believe several should be added to support their content.
2) Recently, genetic variants in the ACSL5 gene, a gene previously associated with rapid weight loss in humans, have been associated with the risk of ALS and a lower fat-free mass in ALS patients (https://doi.org/10.1016/j.celrep.2020.108323). I believe this work to be highly relevant for the topic and should be covered in the review, or at least mentioned in the introduction (paragraph 6, line 62+) when studies that link ALS with energy expenditure, BMI, etc are reported.
Author Response
Thanks for the general comments.
Point to point answers
1) The first part of the introduction lacks appropriate references. Paragraph 1-3 (line 30-45) and 5 (line 54-60) have no references at present and I believe several should be added to support their content.
We apologize for this lack. We have now proceeded to insert appropriate citations.
2) Recently, genetic variants in the ACSL5 gene, a gene previously associated with rapid weight loss in humans, have been associated with the risk of ALS and a lower fat-free mass in ALS patients (https://doi.org/10.1016/j.celrep.2020.108323). I believe this work to be highly relevant for the topic and should be covered in the review, or at least mentioned in the introduction (paragraph 6, line 62+) when studies that link ALS with energy expenditure, BMI, etc are reported.
Thanks for this suggestion. We included this sentence in the introduction section (line 69): Interestingly, variants of the ACSL5 gene, previously associated with rapid weight loss in humans, have recently been associated with ALS risk and lean body mass reduction in ALS patients (22).